# Acylpolyamine Mygalin as a TLR4 Antagonist Based on Molecular Docking and In Vitro Analyses

**DOI:** 10.3390/biom10121624

**Published:** 2020-12-01

**Authors:** Abraham Espinoza-Culupú, Ricardo Vázquez-Ramírez, Mariella Farfán-López, Elizabeth Mendes, Maria Notomi Sato, Pedro Ismael da Silva Junior, Monamaris Marques Borges

**Affiliations:** 1Interunits Graduate Program in Biotechnology, USP/IBu/IPT, São Paulo 01000-000, Brazil; abraham.culupu@usp.br (A.E.-C.); pedro.junior@butantan.gov.br (P.I.d.S.J.); 2Bacteriology Laboratory, Butantan Institute, São Paulo 01000-000, Brazil; abraham.culupu@butantan.gov.br; 3Instituto de Investigaciones Biomédicas, Universidad Nacional Autónoma de México, Ciudad de México 00-16, Mexico; ricardo1v@yahoo.com; 4Microbiology Molecular and Biotechnology Laboratory, Universidad Nacional Mayor de San Marcos, Lima District 15081, Peru; mariella.farfan@unmsm.edu.pe; 5Laboratory of Dermatology and Immunodeficiencies, Medical School, University of São Paulo, São Paulo 01000-000, Brazil; marisato@usp.br; 6Laboratory for Applied Toxinology (LETA), Butantan Institute, São Paulo 01000-000, Brazil

**Keywords:** mygalin, TLR4, inflammation, drug discovery, virtual screening, molecular docking

## Abstract

Toll-like receptors (TLRs) are transmembrane proteins that are key regulators of innate and adaptive immune responses, particularly TLR4, and they have been identified as potential drug targets for the treatment of disease. Several low-molecular-weight compounds are being considered as new drug targets for various applications, including as immune modulators. Mygalin, a 417 Da synthetic bis-acylpolyamine, is an analog of spermidine that has microbicidal activity. In this study, we investigated the effect of mygalin on the innate immune response based on a virtual screening (VS) and molecular docking analysis. Bone marrow-derived macrophages and the cell lines J774A.1 and RAW 264.7 stimulated with lipopolysaccharide (LPS) were used to confirm the data obtained in silico. Virtual screening and molecular docking suggested that mygalin binds to TLR4 via the protein myeloid differentiation factor 2 (MD-2) and LPS. Macrophages stimulated by mygalin plus LPS showed suppressed gene expression of tumor necrosis factor (TNF-α), interleukine 6 (IL-6), cyclooxygenase-2 (COX-2) and inducible nitric oxide synthase (iNOS), as well as inhibition of signaling protein p65 of the nuclear factor κB (NF-κB), resulting in decreased production of nitric oxide (NO) and TNF-α. These results indicate that mygalin has anti-inflammatory potential, being an attractive option to be explored. In addition, we reinforce the importance of virtual screening analysis to assist in the discovery of new drugs.

## 1. Introduction

The main receptors involved in innate immunity to infections are the Toll-like receptors (TLRs), which are expressed on the cell surface (TLR 1, 2, 4, 5, 6, and 10) or on endosomes (TLR 3, 7, 8, and 9) [1]. TLRs are transmembrane proteins that constitute the first line of defense, and they recognize molecular patterns associated with pathogens (PAMPs) that are expressed by infectious agents such as zymosan, peptidoglycan, lipopolysaccharide (LPS), flagellin, and CpG DNA, and mediate the development of an efficient innate immune response system and inflammatory response [2]. These proteins are key molecules in the inflammatory process, and they can be suppressed with specific drugs, antibodies, or inhibitors to treat diseases, such as cancer, sepsis, and asthma; therefore, TLRs are considered a promising future therapeutic strategy for various infectious inflammatory and autoimmune diseases [3,4]. The intracellular activation of TLR4 by LPS requires the cooperation of the adapter molecule MD-2 and cluster of differentiation 14 (CD14) to mediate intracellular transduction signals and initiate a cascade of signal activation coordinated by two distinct transcription factors, the nuclear factor kappa B (NF-kB), responsible for inducing the transcription of the proinflammatory cytokine gene (TNF-α, IL-1, IL-6) and the interferon regulatory factor-3 (IRF3), through the production of type I interferon (IFN-α/β). However, other transcription factors, such as cyclic AMP-responsive element-binding protein (CREB) and activator protein 1 (AP1), are also important [5]. It was demonstrated that TLR4-MD-2 dimers are the basic signaling units for the LPS receptor and that the binding of LPS to the MD-2 molecule results in the formation of a complex, necessary for LPS activity and essential for TLR4 activation. The recognition of LPS by the TLR4 receptor system is accompanied by a series of sequential steps in which the LPS is linked by different LPS-binding proteins and transferred to MD-2/TLR4, then dimerization occurs in the cell membrane, initiating the signaling cascade that leads to the production of pro-inflammatory cytokines and interferons [6].

Proinflammatory mediators, regulated by NF-kB), (COX-2), prostaglandins, and iNOS, are essential to the evaluation of inflammation levels [3]. LPS activates macrophages by binding TLR4, and this activation triggers inflammatory processes through the activation of NF-κB and the synthesis of proinflammatory cytokines [7].

A variety of proinflammatory genes are induced by NF-κB; thus, inhibiting NF-κB activation contributes to the regulation of inflammatory processes and the control of infection. TLR4 is involved in various pathological conditions; therefore, it is considered a potential therapeutic target for the development of drugs [4]. Bacterial infections can cause sepsis by inducing an intense systemic inflammatory process [8]. Thus, molecules that can block the interaction between LPS and TLR4 and its signaling cascade and thereby interference with the inflammatory process must be identified. Several investigations of molecules with LPS antagonist activity, such as curcumin, sulforaphane, xanthohumol, and celastrol [9], and anti-inflammatory potential, such as mygalin, are underway.

Inhibition of TLRs can occur in two ways: By blocking ligand binding to the receptor and by interfering with intracellular signaling pathways to prevent signal transduction [3]. In this context various drugs with pharmacological potential have been developed, such as: The small molecules TAK-242 [10] and ERITORAN [11], which inhibit TLR4; CQ, and CHQ [12,13], which inhibit TLR 7, 8, and 9.

Natural polyamines (putrescine, spermidine, and spermine) are a group of endogenous cationic compounds synthesized by all living cells [14], and depending on the nature of the polyamine, they present several functions, such as anti-inflammatory and immune-suppressive activity [15]. The most studied polyamine is spermine due to its antiproliferative and anti-inflammatory actions on lymphocytes and macrophages [15]. Studies have shown that polyamines are necessary for the expression of TLR2 and that polyamine-induced TLR2 activation plays an important role in regulating epithelial barrier function [16]. Polyamine analogs have been very well studied as anticancer molecules [17,18], although many of their mechanisms of action are unknown. Our molecule named “Mygalin” is a bis-acylpolyamine spermidine analog isolated from the hemocytes of the spider *Acanthoscurria gomesiana*. This molecule has microbicidal effects [19,20], and its invertebrate origin may be responsible for the differentiated characteristics of this molecule.

The integration between biological systems and computational techniques makes it possible to explore drug development and can rapidly provide structural, chemical, and biological data and improve our understanding of potential drug targets. The interactive association between various methods, such as virtual screening and molecular docking, with various immune components has been employed by researchers for the development of pharmacologically active drugs.

In the current study, we investigated the effects of mygalin on TLR4 in LPS-stimulated murine macrophages by in silico analyses of molecular docking and the in vitro production of immune mediators. Our data prove that mygalin can be a promising molecule that efficiently block the LPS/TLR4 signaling at different points.

## 2. Materials and Methods

### 2.1. Reagents

LPS from *E. coli* serotype 0111:B4 as well as, 3-(4,5-dimethylthiazol-2-yl)-2,5-diphenyltetrazolium bromide (MTT), fluoresceine-5-isothiocyanate (FITC), dimethyl sulfoxide (DMSO), α-naphthylethylenediamine (NED), and agarose were purchased from Sigma-Aldrich Chemical Co (St. Louis, MO, USA). RPMI 1640, bovine serum albumin (BSA), antibiotic and primers were purchased from Thermo Scientific (Walthan, MA, USA). The ELISA kit and antibodies were purchased from eBioscience, (San Diego, CA, USA). Mygalin was synthesized and purified at the Center for Research on Toxins, Immune-Response and Cell Signaling (CeTICS), Laboratory for Applied Toxinology (LETA) Butantan Institute following the methodology described by Espinoza-Culupú [19].

### 2.2. Similarity-Based Virtual Screening (VS) of Mygalin

The representation of the mygalin structure in SMILES (Simplified Molecular Input Line Entry) format is as follows: C1=CC(=C(C=C1O)C(=O)NCCCCNCCCNC(=O)C2=C(C=CC(=C2)O)O)O. This structure was used to evaluate the similarity of mygalin with previously reported drugs using the Swiss Similarity [21] web platform against 10,639,400 compounds from the ZINC drug-like database and 19,500 ligands from the Protein Data Bank (PDB) (available 23/02/2020), of PubChem [22] with a Tanimoto threshold ≥ 95% and the ChEMBL database [23] with a Tanimoto threshold ≥ 75.

### 2.3. Ligand Prediction Against Immune Receptors

Mygalin in SMILES format was used to predict its binding to selected immune cell receptors. Mygalin was evaluated by ImmtorLig_DB [24], a repertoire of small molecules that bind to immune system receptors such as TLR 1, 2, 4, and 6, the cytokines IL-1β, IL-4, and IL-6, and major histocompatibility complexes 1 and 2 (MHC-I and MHC-II).

### 2.4. Ligand and TLR4/MD-2 Structures

The molecular structures of mygalin **(**N-[3-({4-[(2,5-dihydroxybenzoyl) amino] butyl}amino) propyl]-2,5-dihydroxybenzamide), curcumin (1,7-bis (4-hydroxy-3-methoxyphenyl) hepta-1,6-diene-3,5-dione) and TAK-242 (ethyl (6*R*)-6-[(2-chloro-4-fluorophenyl) sulfamoyl] cyclohexene-1-carboxylate) were retrieved from the ZINC database [25]. Curcumin and TAK-242 were used as control molecules for the docking study. Initially, the energy profile and geometry optimization calculations were carried out to obtain the lowest energy conformations of three ligands with density functional theory (DFT) using the ωB97X-D function and the 6-31G* basis set level. Selected electronic and physicochemical properties were calculated, including the total energy, solvation energy, dipole moment, electrostatic potential (EP), and molecular volume. All ab initio calculations were performed with Spartan’14 software. The crystal structure of TLR4 and its coreceptor MD-2 was retrieved from the PDB [26] using PDB ID 3VQ1 [27]. TLR4/MD-2 is in a heterotetrameric complex with its LPS (lipid IVa) ligand in two binding pockets. No structural water molecules were found; therefore, all of water molecules, LPS, and other ligands were removed using Discovery Studio Visualizer (DSV) software [28]. The tetrameric receptor was divided into two dimers, namely, TLR4-A/MD-2-C and TLR4-B/MD-2-D, and each complex was used as a rigid structure for independent docking calculations. In addition, the structures of the TLR4/MD-2 complexes were compared based on they superpositions by measuring the root mean square deviation (RMSD) of the protein backbone using SuperPose version 1 (http://superpose.wishartlab.com).

### 2.5. Docking Calculations

The molecular structures of mygalin, curcumin, TAK-242, and TLR4/MD-2 were converted to PDBQT format for docking calculations using AutoDock Tools [29]. All rotatable bonds in each ligand remained free, while the receptor was kept rigid and hydrogen atoms and Gasteiger charges were assigned. In the docking setup, grid dimensions of 36 × 32 × 38 and 36 × 38 × 42 had following coordinates of: x = −6.893, y = 35.973, z = 18.773 and x = 11.774, y = −21.942, z = −2.127, respectively, which centered on MD-2-C and MD-2-D respectively. The spacing box was 1.0 Å, and the exhaustiveness was set to 500. Finally, molecular docking was performed using AutoDock Vina 1.1.2 [30].

### 2.6. Interaction Analysis of the TLR4/MD-2-Ligand Complex

The free energy of the ligand-receptor complexes (TLR4-A/MD-2-C and TLR4-B/MD-2-D) was calculated using the coordinates of the docking calculations.

In the TLR4/MD-2 complexes, the amino acids and molecular interactions that stabilize the TLR4/MD-2-ligand complexes were identified with DSV software. Only the first core of residues belonging to the TLR4/MD-2 binding pocket was included, and the cut off distance between the ligands and the receptor was set at 4 Å for hydrophobic interactions and 3.4 Å for hydrogen bonds.

The interaction energy of the TLR4/MD-2-ligand complex was calculated. The total free energies of interaction included the ligand molecule and all adjacent binding pocket residues. A simplified procedure to calculate the interaction energy (IE) was provided by the following formula: IE = [RL] − [R + L], where IE is the interaction energy of binding, RL is the energy of the complex formed by the receptor residues and the ligand, R is the energy of the TLR4/MD-2 residues, and L is the energy of the ligand. The binding interaction energies were determined by single point calculations using DFT with the ωB97X-D function and the 6-31G* basis set level (for an extensive description of all ab initio methods used in this work, see [31]). Finally, the volume of the binding pocket cavities of each TLR4/MD-2 tetramer was calculated using the Swiss-PDB Viewer program [32]. Docking and ab initio calculations were performed on a 12-core computer with a Xeon processor at 3.4 GHz.

### 2.7. In Vitro Analysis

C57BL/6 mice were euthanized in a CO_2_ chamber, and then the bone marrow from the tibia and femur was extracted to obtain cells that were differentiated into bone marrow-derived macrophages (BMDMs) according to Weischenfeldt [33]. The animals were obtained from the animal center of the School of Medicine of USP and the Butantan Institute. All procedures were performed in accordance with the approval of the Animal Ethics Committee of the Butantan Institute under CEUA protocol no. 1352/14 and CEUA protocol no 5609301018.

### 2.8. Cell Culture and Cytotoxicity Assay

BMDMs, J774A.1 and RAW 264.7 cells were cultivated in RPMI medium supplemented with 10% FBS and gentamicin (25 μg/mL) at 37°C with 5% CO_2_. Macrophages were maintained in cultures with 5% CO_2_ at 37 °C. For the tests, they were plated at a concentration of 1.5 × 10^6^ cells/0.5 mL/well in 24-well plates (Costar, Cambridge, MA, USA). After incubation for 24 h, they were washed with PBS and used for the RT-PCR assay or for the quantification of immune mediators. RAW cells (1 × 10^5^) treated with mygalin (0–2000 µM) for 24 h were tested for cellular viability using the MTT assay (0.5 mg/mL). After 3 h of treatment with MTT, 100 µL of DMSO was added with thorough mixing for 10 min to dissolve the purple formazan crystals [34]. Then, the optical density was measured at 595 nm.

### 2.9. Effect of Mygalin on LPS-Stimulated Cells

Macrophage cell lines or BMDMs were plated at a concentration of 1.5 × 10^6^ cells/0.5 mL/well in 24-well plates (Costar) for 24 h at 37 °C. After washing with PBS, the cells were incubated or not with mygalin (50, 150, or 450 μM) for 1 h before the addition of TLR4 agonist (LPS 100 ng/mL) for 6 h. After washing with PBS, the cells were harvested for total RNA extraction, and after 24 h of stimulation, the culture supernatants were collected for cytokine and (NO) measurements.

In some assays after 1 h of treatment of J774A.1 cells with mygalin (50 and 150 µM), the culture medium was removed, the cells were washed with medium new medium containing LPS (100 ng/mL) was added and then the cells were incubated for 24 h to confirm the influence of mygalin on LPS-induced NO production.

### 2.10. RNA Isolation and RT-PCR

After 6 h of treatment, the cells were washed with PBS (500 μL per well). Total RNA was extracted from the macrophages with 180 μL of QIAzol lysis reagent (Qiagen^®^) and purified using a Direct-zol kit (ZYMO RESEARCH) following the manufacturer’s instructions. RNA was quantified by measuring the absorbance at 260 nm using a GE NanoVue Plus^TM^ spectrophotometer (GE Lifesciences) and the integrity was visualized on 1.2% agarose gels stained with red GelRed^TM^ (Biotium) [35]. Total RNA (100 ng) was converted into cDNA using a High Capacity RNA-to-cDNA kit (Applied Biosystems^TM^). For the negative control, the enzyme was replaced with water. cDNAs were used as a template for the amplification of the genes o-f interest using specific primers designed in this study (Table 1) with Platinum^®^ Taq DNA Polymerase (Invitrogen), and the final reaction volume was 25 µL. All PCRs included the negative reverse transcriptase (RT) and negative PCR control. RT-PCR products (10 µL) were separated by electrophoresis on 1.5% agarose gels at 70 V in TAE 1X buffer and stained with GelRed, and bands were visualized using an electronic photo-document system (UVITEC, Cambridge). The densities of the PCR products were quantified using the ImageJ program (http://imagej.nih.gov/ij/) and compared against the expression of the β-actin gene.

### 2.11. Measurement of NO

Supernatants of BMDMs, J774A.1 or RAW cells were collected after treatment for 24 h to measure the presence of nitric oxide, which was quantified using the Griess reaction method [36]. To assess the presence of nitrite (NO_2_^−^), which is a common reaction product, 50 μL of the supernatant and 50 μL of a 1% sulfanilamide solution were added to a 96-well plate and incubated for 10 min at room temperature under dark conditions. Then 50 μL of 0.1% NED was added followed by further incubation under in the same conditions. The absorbance was then measured at 550 nm. The NO_2_^−^ concentration in each sample was based on the standard sodium nitrite (NaNO_2_) curve constructed between 1 and 100 μM. The tests were performed in quadruplicate (N = 4).

### 2.12. Neutralization Assay of LPS by Mygalin

LPS (100 ng/mL) was treated with mygalin (50, 150, or 450 μM) for 45 min in a water bath at 37 °C. The mixture was used to stimulate macrophages for 24 h, and the supernatants were collected to measure the NO production. Polymyxin (30 μg/mL) was used under the same conditions as a positive control to block the biological effects of LPS [37].

### 2.13. Conjugation of FITC-LPS

In this assay, 1 mg of LPS from *E. coli* serotype 0111: B4 was conjugated with 4 mg of FITC following the methodology of Skelly [38] with minor modifications. Briefly, the proportion was 1:4 in 1.5 mL of sodium borate (0.1 M) pH 10.5 for 3 h at 37 °C and 180 rpm. The mixture was dialyzed by membrane dialysis with a MWCO of 1000 (Spectra) overnight against 0.15 M NaCl with 4 exchanges. All solutions were prepared with sterilized ultrapure water.

### 2.14. Interaction of Mygalin with FITC-LPS

The interaction between mygalin and LPS-FITC was studied by monitoring the fluorescence of conjugated FITC-LPS. For this analysis 10 µg/mL conjugate was incubated with different concentrations of mygalin (50, 150, or 450 µM) in 100 µL of saline solution using black 96 well plates protected from light. The plates were incubated for 45 min at 37 °C to measure the fluorescence changes, which were calculated using 485/535 nm excitation and fixed emission on a PerkinElmer Victor 3™ 1420 multilabel counter fluorometer and another test at 475 nm excitation and 500–560 nm emission as previously described [39] using a Cytation 3 Imaging Reader Biotek Instrument. All experiments were performed in quadruplicate.

### 2.15. Measurement of TNF-α

The presence of TNF-α was analyzed in the supernatants of the cell cultures using ELISA immunoenzymatic assay kits from eBioscience following the manufacturer’s guidelines. The reaction was measured at 450 nm in a microplate reader (Multiskan EX, Primary EIA). The amount of cytokines present in the samples was calculated based on standard curves.

### 2.16. Western Blot Analysis

Cells were harvested and lysed with RIPA buffer (150 mM NaCl, 1% Triton X-100, 0.5% sodium deoxycholate, 0.1% SDS, 50 mM Tris-HCl pH 8, protease inhibitor) for 30 min at 4 °C following the manufacturer’s instructions. Cell lysates were centrifuged at 14,000 rpm for 20 min at 4°C, and the protein concentration was measured using a Pierce ™ BCA protein kit (Thermo Scientific). The total proteins (30–50 µg) were separated on 10% SDS-PAGE gels. The proteins were transferred to a nitrocellulose membrane and preincubated for 2 h at room temperature in saline solution buffered with Tris (pH 7.5) containing 0.1% Tween-20, 5% skim milk, and specific antibodies (iNOS, β-actin or NF-κB/p65) at 1:1500 with gentle agitation overnight at 4 °C. Anti-rabbit IgG antibody conjugated to horseradish peroxidase was used as the secondary antibody at a dilution of 1:2000. Protein bands were detected by chemiluminescence enhanced with 1X SignalFire ™ ECL reagent and photographed using an electronic documentation system (UVITEC, Cambridge).

For the NF-κB/p65 assays, RAW cells (2 × 10^6^ cells/well) were pretreated with mygalin (50, 150, or 450 µM) for 1 h and then incubated with LPS (100 ng/mL) for 30 min. Nuclear and cytoplasmic proteins [40] were extracted with RIPA buffer, and Western blotting was performed using a specific antibody against NF-κB/p65 as described above.

### 2.17. Statistical Analysis

All results were analyzed using Student’s *t*-test and one-way ANOVA, and the difference between groups were determined by Dunnett’s multiple comparisons test and analyzed by the program GraphPad Prism 7 (Graph Pad, San Diego, CA, USA). The data were considered statistically significant at *p* < 0.05, and the results represent the mean and standard error of the mean (±SEM) from at least three independent experiments.

## 3. Results

### 3.1. Virtual Screening

Mygalin in SMILES format was used to perform virtual screening with different databases using web servers, to identify molecules with a Tanimoto similarity index > 75% and different biological activities such as bactericidal [41,42], siderophoric [43], and antimalarial activity [44] as shown in Appendix A. The results indicate that mygalin could have more than one biological activity

### 3.2. Prediction Against Immune Receptors

Mygalin was evaluated in the virtual repertoire of small molecules against immune receptors ImmtorLig_DB, in which the program database counts human immune receptors such as Toll-like receptors (TLR1/TLR2, TLR4/MD-2, TLR2/TLR6) and mincle; MHC-I and MHC-II; costimulatory molecules CD28, CD40, CD80, and CD86; coinhibitory molecules CTLA-4, PD-L1, Tim-3, decoy receptor, Fas ligand, and Fas receptor; cytokines IL-1β, IL-2, IL-4, IL-6, IL-17, and IL-23; and cell adhesion molecules ICAM, VCAM, CEACAM1. Mygalin was evaluated with all of these receptors, and it appears that mygalin could have binding affinity for IL-4 (score of 0.161), TLR2/TLR6 (0.153), TLR4/MD-2 (0.136) and other molecules as shown in Table 2. These results indicate that mygalin could act as a binding molecule for some TLR 2, 4 and 6) as a signaling or blocking molecule. The in silico analysis suggests that mygalin has affinity for receptors or molecules that participate in the control of the immune response and could have biological activities that are different from the bactericidal and siderophoric activities found in our previous study [19].

### 3.3. Molecular Structure of the Ligands

The electronic and physicochemical properties of mygalin, curcumin, and TAK-242 were obtained by ab initio calculations. The compounds exhibited different EP patterns (Figure 1). Mygalin and curcumin were more similar to each other than to TAK-242 and showed negative regions (red) exposed at the ends of the structure that are capable of forming hydrogen bonds and electrostatic interactions. Because of the compact structure of TAK-242 it had a very concentrated negative region at the center. The three ligands were polar molecules that showed large dipole moments (especially curcumin and TAK-242), while mygalin had a smaller dipole moment due to its symmetry (Figure 1).

Regarding the physicochemical properties, the solvation energies and polar surface showed that mygalin is the most water-soluble compound and TAK-242 the least water-soluble. Finally, the greater volumes of mygalin and curcumin compared to TAK-242 could mean improved occupation of a large binding pocket (Table 3).

### 3.4. Structural Comparison between TLR4/MD-2 Complexes

The X-ray structure of the complexes (TLR4-A/MD-2-C and TLR4-B/MD-2-D) without ligands was compared by three-dimensional superposition and RMSD. The RMSD of 0.225 Å and superposition showed a good overall structural agreement between the two complexes. However, small conformational differences were observed in the side chains of the residues between two complexes. In the LPS-binding domain, most of residues showed a similar conformation or small differences, whereas W23, S47, and I153 (MD-2) showed highest conformational differences (Appendix A).

### 3.5. Docking Calculations in the TLR4/MD-2 Complex

In the receptor-ligand complexes obtained from the docking calculations, the free energies of interaction were analyzed based on quantum mechanics theory to obtain more accurate results than the docking affinities resulting from AutoDock Vina. The free energy calculations were analyzed. This assessment investigated the TLR4/MD-2 complexed with ligands obtained from docking calculations.

As a result of the docking calculations, 9 binding receptor-ligand modes were obtained, and only the lowest free energies of the three ligands for TLR4-A/MD-2-C and TLR4-B/MD-2-D were considered for further analysis.

The MD-2 residues bound to the ligands showing hydrophilic (polar) interactions reached a distance of approximately 3.4 Å, while those with hydrophobic interactions (van der Waals) reached 4 Å (Figure 2). The number of binding residues was identified, all of which belonged to MD-2. Most of the residues were hydrophobic (7–15), while the number of polar residues ranged from 2 to 4, and interestingly, R90 was found to have polar interactions in all the low free energy complexes. R90 is on the border of MD-2 close to the multimerization interface and established strong polar interactions with the oxygens of the three ligands. Regarding the free energy of interaction, the lowest energy conformation of each of the three ligands was similar in both binding pockets of MD-2 (C and D), with mygalin showing the lowest free energy of interaction of the three molecules (Table 4). Mygalin and curcumin showed extended conformations along the MD-2 binding pocket and anchored to R90 through their phenolic hydroxyl, while TAK-242 associates with R90 through its sulfonamide oxygen (Figure 2).

### 3.6. In Vitro Cytotoxic Activity

After the in silico analysis of mygalin binding to LPS, we performed an in vitro evaluation to verify whether TLR4 binding to LPS may influence the activation of the signaling pathway. First, we analyzed the effects of mygalin on macrophage viability. The effects of mygalin (Figure 1A) on the viability of mouse BMDMs and the cell lines J774A.1 and RAW 264.7 were assessed using the MTT assay. No morphological alterations were observed, and 100% viability of the cells was observed after treatment with mygalin. These data show that mygalin is not toxic to macrophages at the concentrations evaluated (Figure 3).

### 3.7. Mygalin Attenuates LPS-Induced iNOS, TNF-α, IL-6 and Cox-2 mRNA Expression in RAW Cells

The mRNA expression of iNOS, TNF-α, COX-2, and IL-6 was analyzed by RT-PCR (Figure 4) after 6 h of treatment with LPS. Treatment of cells with mygalin at different concentrations reduced transcriptional expression in a dose-dependent manner. Inhibition of gene expression started at 150 µM mygalin and was more pronounced at 450 µM except for IL-6, for which 50 µM had a significant effect (Figure 4). TAK-242, a signal inhibitor via TLR4, was used as a control. The results indicated that mygalin was able to interfere with LPS-induced iNOS, TNF-α, IL-6, and COX-2 expression levels and, decreased the inflammatory effects generated by the TLR4 agonist.

### 3.8. Effects of Mygalin on the Production of TNF-α

We next evaluated the effects of mygalin on TNF-α production. Figure 5 shows that mygalin at concentrations of 50, 150, and 450 µM in LPS-activated macrophage cultures significantly reduced TNF-α production in a dose-dependent manner. At higher dose treatments, the levels of this cytokine approached those of the control TAK-2, which inhibits TLR4. The same effect has been observed for cordycepin (a nucleoside analog), curcumin, and spermidine after the cells were stimulated with LPS [40,45].

### 3.9. Effects of Mygalin on NO Production in Macrophages

In another assay, after 1 h of pre-treatment with mygalin (50 and 150 µM); J774A.1 cells were washed with medium and new medium containing LPS (100 ng/mL) was added. NO was measured in the culture supernatant after 24 h. The data shown in Figure 6, suggest that even in the absence of mygalin at the time of the addition of LPS, the anti-inflammatory effect of this molecule was maintained. There was a 40% reduction in NO production by J774A.1 cells, similar to that obtained with BMDMs (40%) and RAW cells (48%) treated with the same concentration, when both molecules were present (Figure 7a and b).

In another assay, the levels of NO in two cell types, BMDMs and RAW cells, were quantified and correlated with the previously observed gene expression shown in Figure 4a,b. Figure 7 shows that mygalin in the macrophage cultures at concentrations of 50, 150, and 450 µM did not generate significant amounts of NO. However, when mygalin was added to the LPS-activated cultures, the level of NO was significantly reduced in a dose-dependent manner. The most pronounced effect occurred when we used 450 µM mygalin, thus confirming the previously observed reduction in gene expression.

### 3.10. Neutralizing LPS with Mygalin

Mygalin is able to interact with LPS; therefore, we analyzed whether this complex interaction may block the interaction with TLR4. Mygalin at different concentrations (50, 150, and 450 µM) was incubated with a fixed concentration of LPS (100 ng/mL) for 45 min at 37 °C. Then, this solution was added to the cell cultures for 24 h, and NO production was measured by the Griess reaction (Figure 8). The results indicated that mygalin was able to neutralize LPS activity, thereby reducing NO production in a dose-dependent manner. The most accentuated effects were 150 and 450 µM mygalin. The maximum neutralization effect was close to the control, with polymyxin B showing intense inhibitory activity of the action of this endotoxin.

### 3.11. Binding of Mygalin to LPS-FITC

To demonstrate the mygalin-neutralizing effects of LPS, its binding with the endotoxin was quantified. Mygalin was incubated at several concentrations with LPS conjugated to fluorescein. Fluorescein reduction was monitored and compared to conjugated LPS. After incubation of mygalin with different doses of LPS labeled with FITC, we observed a decrease in fluorescence compared to the control without mygalin. The decrease in fluorescence was proportional to the increase in the concentration of mygalin. Similar results were found with peptides after incubation with LPS, thus showing the interaction between the two [39]. Our data confirm that mygalin can interact with LPS (Figure 9).

### 3.12. Mygalin Suppresses the Inflammatory Response of LPS by Interfering with the Expression of iNOS and NF-Κb p65 in RAW 264.7 Macrophages

TLRs recognize molecular patterns expressed by different infectious agents, which leads to activation of the transcription factors NF-κB and IRF [46]. TLR4 recognizes LPS triggered cellular signaling processes. Therefore, the effects of mygalin on NF-κB expression, the main transcription factor that regulates the synthesis of inflammatory proteins, was analyzed. In parallel, we evaluated the expression of the iNOS enzyme that controls NO synthesis. Figure 10 shows that stimulation of cells with LPS (100 ng/mL) upregulates the expression of both iNOS and NF-κB p65, while mygalin (50, 150, and 450 µM), was able to reduce this response after 30 min of treatment. This result indicates that mygalin could block LPS signaling, thereby decreasing the inflammatory effect induced by TLR4 agonists.

## 4. Discussion

Virtual screening (VS) for drug discovery is becoming an essential in silico tool to predict new biological activities of small molecules or existing molecules [47,48,49]. Online tools or web servers, such as Swiss Similarity, or databases of chemical compounds, such as PubChem and ChEMBL, facilitate these types of analyses and allow comparisons of the structures of interest with existing structures in the databases. In this study we evaluated the molecule mygalin (Figure 1a) with these databases and identified molecules with structural similarity and known activities (Appendix A). Among the molecules with a Tanimoto index > 75% included a polyamine analog with antimalarial activity [44], with a similarity of 91.3%; molecules with activity against viruses, such as HCV [50] and HIV [51]; a catechol-O-methyltransferase (COMT) inhibitor [52]; and molecules with siderophoric [43], antibacterial [41,42], and anticancer [53,54] activities. These last three biological activities have been demonstrated in in vitro and in vivo tests, with the antibacterial activity reported by Pereira [20] and Espinoza-Culupú, who showed the molecular mechanism of action in bacteria and their siderophoric activity [19]. In the case of anticancer activity, silver nanoparticles conjugated with mygalin [55] showed intense cytotoxic activity against cancer cell lines, thus confirming the in silico predictions. In addition, it was predicted that mygalin may have other properties in nervous system, antiviral and COMT inhibitory activity.

Another analysis carried out with mygalin predicted whether it could interact with cellular receptors related to immune cells. For this, mygalin in SMILES format, which was evaluated in ImmtorLig_DB, was found to bind to TLRs, including TLR2 and TLR4, as observed in Table 2. This database includes small drugs with activity against these receptors and could be used in comparisons with molecules such as ours. We obtained scores that indicated possible binders for mygalin. Through these VS approaches, different TLR antagonist molecules have been identified included molecules that target TLR2 [56] and TLR4 as described by Urban [57], who found small molecule TLR4 antagonists using the molecular docking and experimental testing methodologies and validated the results of VS.

In silico and in vitro studies have been carried out to search for and verify TLR-binding molecules, including TLR4, thus providing information on their activation or inactivation mechanisms. The key interactions in this process allow for the discovery of new molecules as modulators. The VS results for mygalin indicated that it was a possible binding agent for TLR4; therefore, we decided to perform a molecular docking study to corroborate these predictions, and we included two drugs with scientific evidence of antagonistic activity against TLR4: Curcumin and TAK-242 [10].

Before the docking study, the X-ray structures of the two TLR4/MD-2 complexes were compared through RMSD and three-dimensional superposition. The RMSD of 0.225 Å indicated structural similarity between the two complexes. The superposition showed small differences in the LPS-binding pocket residues, especially in W23, S47 and I153 (MD-2); however, most of residues showed good structural agreement (Appendix A). These small differences in the conformation of the residues could be correlated with the difference in the volume of the binding pocket (Table 3), number of residues bound to ligands and free energies of interaction (Table 4). However, the overall activity between the three ligands was preserved.

A docking study was performed to predict the binding modes and affinities of mygalin, curcumin and TAK-242 with TLR4/MD-2, which focused on the LPS binding domain.

Molecular interactions were analyzed in all receptor-ligand complexes and the binding domains showed conserved residues and a similar pattern of interactions in both TLR4/MD-2 complexes. Affinities were obtained from docking calculations, but the energy differences between the ligands were less than 3 kJ/mol. Due to such small differences in affinities and to improve accuracy, free energies of interaction were evaluated with theory of quantum mechanical (Table 4).

The binding of agonistic ligands has been proposed to cause dimerization of the extracellular domains, which is believed to trigger the recruitment of specific adaptor proteins to the intracellular domains, thus initiating a signaling cascade [5]. LPS binds to this pocket and directly mediates dimerization of the two TLR4/MD-2 complexes [27]; thus LPS is expected to bind to a dimer (TLR4-A/MD-2-C) and multimerization subsequently occurs with the coupling of another dimer (TLR4-B/MD-2-D). The LPS binding site is a large hydrophobic pocket in MD-2 and a cluster of polar and positively charged residues of TLR4 and MD-2 that directly bridges the two components of the multimer [5]. Although mygalin is a polar compound and highly water soluble, as indicated by the physicochemical and electronic properties (Table 3), the free energy of interaction shows that mygalin can form TLR4/MD-2 complexes in the dimeric configuration in the same way as other antagonists (curcumin and TAK-242) and with lower energies in the dimeric complexes (Table 4).

Synthetic small antagonists have been shown to bind to TLR8, stabilize the dimer conformation and prevent further conformational changes that are necessary for TLR8 activation but to antagonize the binding of TLR8 activators that occupy the active site [58]. The docking study showed that mygalin was capable of partially occupying the binding pocket of MD-2 (Figure 2), which is due to the small size compared to LPS, and it interacts with some residues that are used by LPS during its binding and part of the dimerization interface, such as I80, R90, F126,Y131 of MD-2 Y131, or MD-2 [5,27,59].

The binding pocket is a hydrophobic cavity centered on MD-2, although it has a polar patch on the edge of MD-2 near to the dimerization area of TLR4 [5]. In the TLR4/MD-2–ligand interaction, the driving force is to expel the ligands from the bottom of the binding pocket and to occupy the polar patch, as occurred with mygalin by forming a hydrogen bond with R90. However, the absence of a group with a negative formal charge reduces the possibility of a very strong ionic association with the positively charged residues of TLR4 and may be unable to cause a conformational change to occur, as has been shown with other ligands that lack phosphate groups [5]. Mygalin had very good energy agreement to establish a stable complex with MD-2. Due to its polar and water-soluble characteristics, mygalin was able to interact with a patch of polar residues at the dimerization interface but lacked groups capable of forming ionic interactions with TLR4, thereby may is insufficient to generate activation or multimerization as has been proposed for other synthetic antagonists [60].

Since the in-silico results predicted mygalin to be a possible binder of TLR4, we decided to carry out in vitro analyses to confirm these results. First, we evaluated the cellular viability after treatment with mygalin in macrophages. Previous studies from our laboratory showed that low concentrations of mygalin (0–100 µM) were not toxic to splenocytes [61]. In our tests, we used higher concentrations of mygalin, with a maximum of 2000 µM, and tested two other cell types, primary cells derived from BMDMs and the cell lines RAW and J774A.1 (data not shown). No change in the viability of these cells was proven (Figure 3). These results confirm that even at a high dose, mygalin is not toxic to eukaryotic cells. Based on these data, we chose three doses (50, 150, and 450 µM) of the drug for our tests at levels below the levels required for biological activity of known polyamines, which are in the millimolar range [61]. These data showed that the effects of mygalin in mammalian cells differs from those found in bacteria, where doses greater than 1000 µM are toxic to *E. coli* (Espinoza-Culupú, 2019); these effects are attributed to differences in the membrane structure between these types of cells.

Inflammation is a complex reaction that is common to various injuries in the body and usually presents itself as a protective mechanism. It is a cascade reaction involving several inflammatory mediators, including NO, PGE2, TNF-α, IL-6, IL-1β, and IL-10 [62]. Because mygalin acts mainly in Gram-negative bacteria [20] and LPS is the main inflammatory component of these bacteria, we investigated the potential inflammatory mechanism of mygalin in LPS-stimulated phagocytic cells. For this, a series of assays were performed to determine the mediators, such as TNF-α and IL-6, and the gene expression of the enzymes iNOS and COX-2 to correlate with the production of NO and the expression of NF-κB p65 which are pathways related to inflammation.

Pretreatment of the cell cultures with mygalin reduced the expression of the iNOS, TNF-α, COX-2 and IL-6 genes after 6 h of stimulation with LPS. The cellular effects were more significant after treatment with higher concentrations of the drug (Figure 4). We noticed that the concentration of 450 µM had a similar effect to TAK-242, which is a potent inhibitor of cytokines and signals via TLR4. Similar results regarding the expression of the same genes against stimulation by LPS have been described for other molecules, such as β-carotenes [63], resveratrol [64], and curcumin. These data suggest that mygalin can act via TLR4, which was similar to the TAK-242 molecule that binds to the TLR4 receptor, blocking LPS actions.

We next assessed the effects of mygalin on the levels of TNF-α and NO in the culture supernatant after 24 h of stimulation with LPS, and the cell lysate was used for protein analysis by Western blot. A great reduction in the synthesis of TNF-α (Figure 5) and NO (Figure 6, Figure 7 and Figure 8) was observed, which was similar to the reduction observed for the gene expression of these mediators, and a reduction in the enzymes iNOS and COX-2 (Figure 4), indicating that mygalin plays an anti-inflammatory role during LPS activation. The data in Figure 6 showed that even in the absence of mygalin during stimulation with LPS, the anti-inflammatory effect of the molecule remained, reducing NO production, similarly to the observed effect when the molecules added together. Our data showed that mygalin can bind to LPS neutralizing its activity, as well as bind to the adapter protein MD-2 inhibiting intracellular signaling pathways that lead to reduced LPS activity. This suggests that the pretreatment of cells with mygalin was sufficient to block signaling pathways induced by LPS activation. The level of NO reduction during the treatment of J774A.1 cell with 150 µM of mygalin was similar to that observed in BMDMs and RAW cells stimulated with both molecules, ranging from 40 to 48% (Figure 7a,b). These results indicate that mygalin can bind to TLR4 as demonstrated by in silico and molecular docking assays.

In addition to this, there are others molecular signals that can occur, due to phosphorylation of intracellular proteins, which reflect in the cascade of signals generated during the contact of LPS with TLR4, which were not analyzed here, but are being investigated in our laboratory.

TNF-α [65] is one of the main mediators of the inflammatory response, and it is activated by molecular signals mediated by the contact of PAMPs with TLRs. LPS caused an increase in the levels of TNF-α, while treatment of the cells with mygalin reduced the expression of TNF-α mRNA, reflecting inhibition of TNF-α synthesis in a similar way to the control drug TAK-242. These results are similar to those described in the literature, where molecules such as spermidine [45], curcumin [66], and peptides [67] were shown to inhibit the synthesis of TNF-α and other proinflammatory cytokines.

The production of TNF-α is essential for the synergistic induction of NO synthesis both in vitro and in vivo [68]. Bacterial products, activate iNOS, which generates high concentrations of NO; however, overproduction of NO can be toxic. Changes in NO levels through the induction or inhibition of iNOS enzymatic activity represent a method of assessing the effects of a drug on the inflammatory process. Our data indicated that both primary culture cells and cell lines treated with mygalin had the same NO reduction pattern, showing that this effect does not depend on the cell type (Figure 7).

Burns [69], combined several spermine analogs with fluorescein-labeled lysine and observed that they functioned as a powerful LPS sequestrators. The fluorescence intensity of these analogs, as well as the NO produced by the J774A.1 macrophage, proved that the analogs neutralized the toxicity of LPS. The same author [70] also created a library of compounds, some with an acyl group and noted that these compounds neutralized LPS. To determine whether mygalin specifically binds to LPS and neutralizes its activity, the molecules were preincubated together, and then the mixture was subsequently used to activate cells. Preincubation of these molecules dramatically reduced NO production in a similar manner to that of polymyxin B (Figure 8). This finding corroborates our previous results. Mygalin, as well as other compounds, such as antimicrobial peptides [71] and the polyamines spermidine at a concentration of 800 µM [72], may act to neutralize the activity of LPS.

Another complementary approach was to conjugate LPS with FITC and measure this interaction with a fluorescence assay. Our results (Figure 9) showed that mygalin interacts with the labeled LPS and, the fluorescence intensity decreased as the mygalin concentration increased. The same approach was used by another author with kukoamine B [73], which interfered with the binding of LPS with TLR4. Our data prove that mygalin is a TLR4 antagonist.

The binding of LPS to TLR4 activates inflammatory gene transcription pathways, and the production of PGE2 and NO through the activation of the enzymes COX-2 and iNOS [74]. NF-κB plays a very important role in the regulation of the immune response mediated by different factors such as stress, antigens or diseases with an inflammatory response [75], thereby inducing the expression of several proinflammatory [76] and anti-inflammatory [77] genes. We observed that the treatment of cells with mygalin (450 µM) and LPS significantly reduced the expression of NF-κB p65 and iNOS (Figure 10), similar to level observed with our control drug TAK-242. Therefore, mygalin suppresses NO, TNF-α, and COX-2 synthesis by regulating NF-κB, preventing its translocation and inhibiting the inflammatory cascade induced by LPS. These findings suggest that mygalin can block activation by LPS via TLR4 as with other anti-inflammatory molecules, such as corticosteroids [78], peptides [79], and chalcones (including okanin [80] and others thereby showing that thispathway has therapeutic potential for different inflammatory diseases and cancers) [80].

## 5. Conclusions

In conclusion, in this study VS of mygalin similarity was shown to be a fast and economical method of finding potential targets. We demonstrated for the first time via in silico analysis, VS, molecular docking, and in vitro analysis of LPS inflammation in murine macrophages that acylpolyamine mygalin is a TLR4 antagonist since it sequesters LPS and suppresses the gene and protein expression of TNF-α, IL-6, iNOS, and COX-2 via inhibition of NF-κB p65 protein signaling. Together, our data showed that mygalin can bind to LPS, as well as to the adapter protein MD-2, inhibiting intracellular signaling pathways that lead to the reduction of LPS-induced inflammation, and that mygalin may be an attractive option to protect against inflammatory disorders. Finally, our research provides a model for identifying specific molecules with anti-inflammatory potential, using online tools through in silico analysis and an in vitro model suggesting that MD-2 may be one of mygalin target.

Therefore, we suggest that the mygalin action model represented in Figure 11, in which mygalin blocks the inflammatory effects of LPS via TLR4.

## Figures and Tables

**Figure 1 biomolecules-10-01624-f001:**
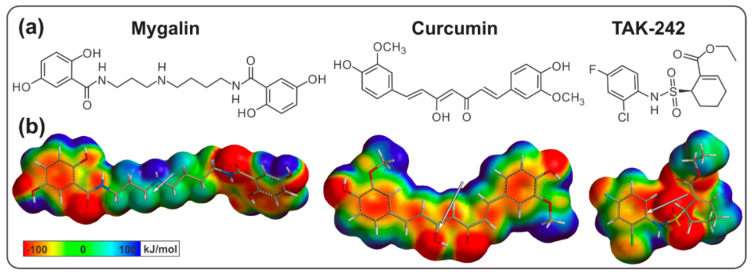
(**a**) Molecular structure of the ligands constructed in the program ChemDraw. (**b**) Electronic structure of the ligands in their lowest energy conformations, with curcumin in its enolic form. Electrostatic potentials and dipole moment vector (arrow) are shown.

**Figure 2 biomolecules-10-01624-f002:**
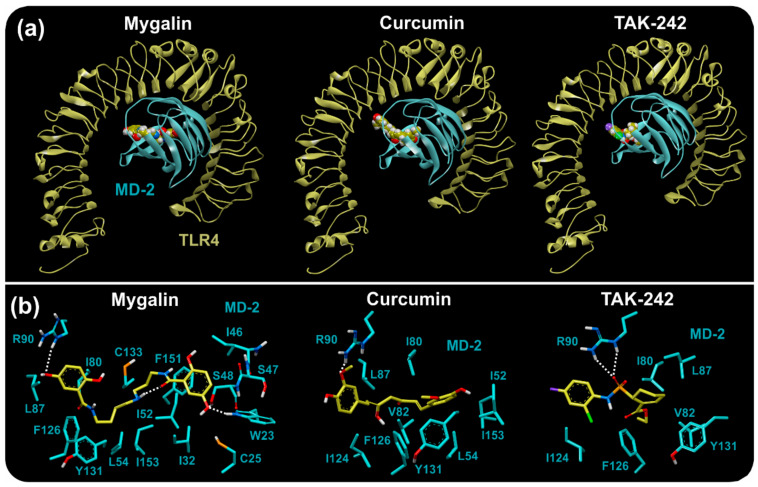
TLR4/myeloid differentiation factor 2 (MD-2) complexes from the docking study of three ligands. Representative MD-2-D binding pocket of the TLR4/MD-2 complexes; the same event occurred in MD-2-C. (**a**) TLR4/MD-2 complexes showed a potential center binding site in MD-2. Mygalin in the MD-2 binding pocket showed no interaction with TLR4 residues. (**b**) Ligands (yellow) upon interaction with MD-2 (blue) residues. Most of the interactions were hydrophobic, although residues R90 and W23 show polar interactions, due to hydrogen bonds. Dotted lines show hydrogen bonds.

**Figure 3 biomolecules-10-01624-f003:**
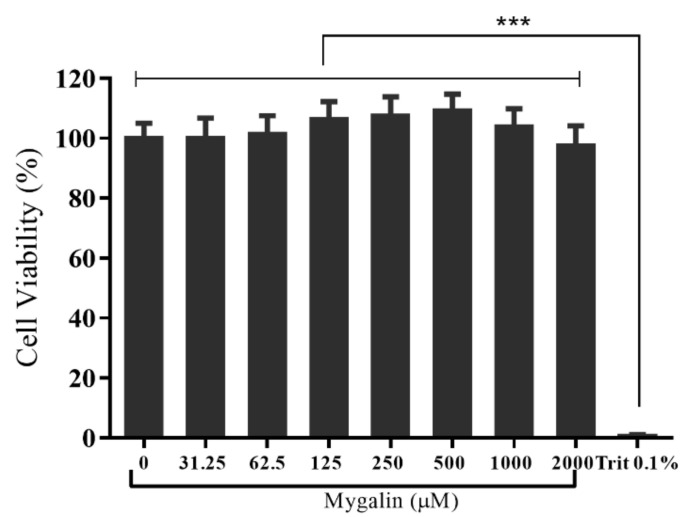
Effect of mygalin on cell viability. The effects of mygalin (0–2000 µM) were compared with the effects of 0.1% Triton (nonviable cells). The results represent the mean ±SEM from three independent experiments (*** *p* < 0.001).

**Figure 4 biomolecules-10-01624-f004:**
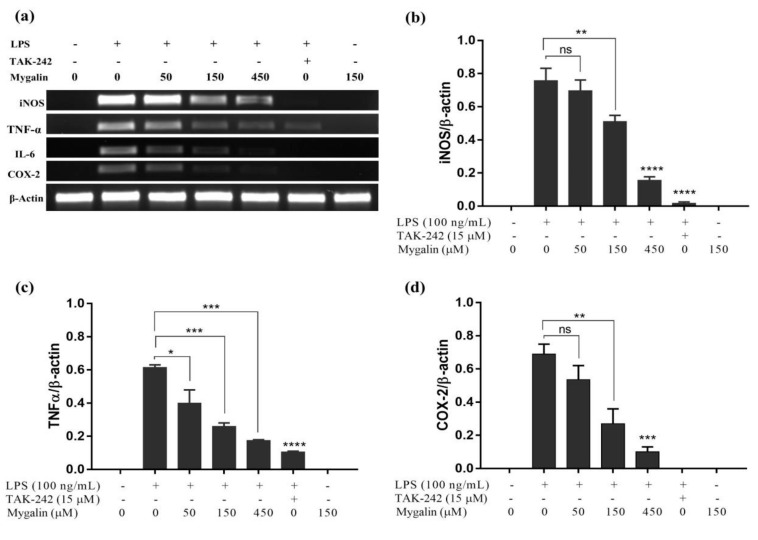
Mygalin inhibits the mRNA expression levels of iNOS, TNF-α, and COX-2 in RAW cells stimulated with LPS. Cells were pretreated with mygalin for 1 h and then stimulated with LPS for 6 h. (**a**) Total RNA was isolated from the cells treated with LPS and submitted to RT-PCR. The agarose gels were stained with GelRed and visualized. PCR products were quantified by densitometry. Relative levels of (**b**) iNOS, (**c**) TNF-α, and (**d**) COX-2 mRNA were normalized against the β-actin gene and compared with the group treated with LPS. The bars represent the mean ± SEM of the relative amounts of mRNA from the genes evaluated from three independent experiments. (* *p* < 0.05, ** *p* < 0.01, *** *p* < 0.001, **** *p* < 0.0001 and ns: Not significant). RT-PCR: Reverse transcription followed by polymerase chain reaction. TAK-242 was used as an LPS inhibitor control.

**Figure 5 biomolecules-10-01624-f005:**
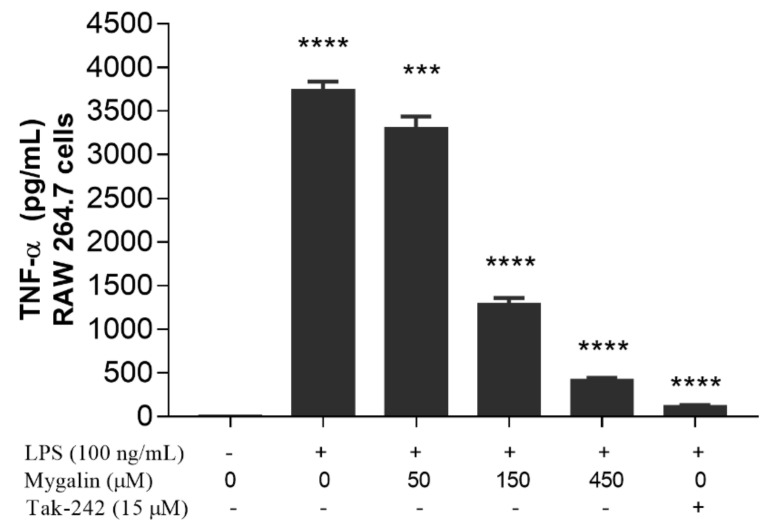
Effects of mygalin on TNF-α levels in cells stimulated with lipopolysaccharide (LPS). The levels of TNF-α secreted in RAW cells were measured 24 h after stimulation with LPS. The cells were pretreated for 1 h with mygalin, and TAK-242 was used as a positive control as an LPS blockade. The data represent the mean ± SEM of three independent experiments carried out in triplicate and the *p* value was calculated using multiplex comparisons with the Tukey test (*** *p* < 0.001, **** *p* < 0.0001 compared to LPS).

**Figure 6 biomolecules-10-01624-f006:**
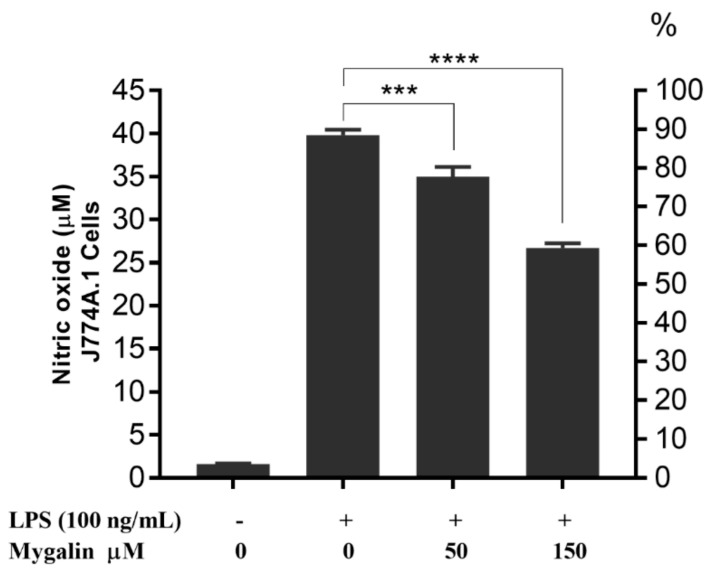
Effect of mygalin on the production of NO induced by LPS in J774A.1 cell. Cells were pretreated for 1 h with mygalin (50 and 150 µM), and then washed with medium and new media plus LPS (100 ng/mL) were added. After 24 h NO was measured in the culture supernatant The bars represent the mean ± SEM of three independent experiments. (*** *p* < 0.001, **** *p* < 0.0001).

**Figure 7 biomolecules-10-01624-f007:**
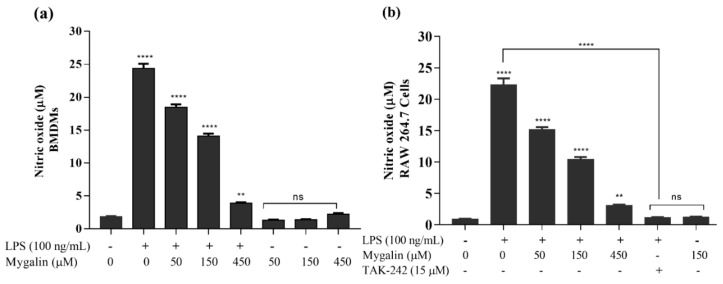
Effects of mygalin on the production of nitric oxide (NO) induced by LPS in macrophages. The cells were pretreated for 1 h with mygalin (50,150 or 450 µM), and then LPS was added. The culture supernatant was obtained after 24 h to quantify the NO level. (**a**) Bone marrow-derived macrophages (BMDMs) and (**b**) RAW 264.7 cells. The data represent the mean ± SEM of three independent experiments performed in triplicate (** *p* < 0.01, **** *p* < 0.0001 compared to LPS) and ns (not significant) compared to the untreated group.

**Figure 8 biomolecules-10-01624-f008:**
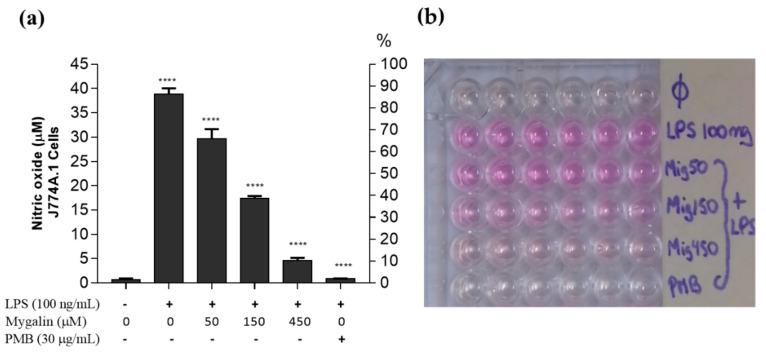
Mygalin neutralizes LPS activity. LPS was incubated with mygalin (50, 150, and 450 µM) or polymyxin B (PMB, 30 µg/mL) for 45 min at 37 °C and then added to the J774A.1 macrophage cell culture. (**a**) NO_2_^−^ was quantified at 550 nm after 24 h. (**b**) Visualization of the Griess reaction. The data represent the mean ± SEM of three independent experiments performed in triplicate (**** *p* < 0.0001 compared to LPS) and ns (not significant) compared to an untreated group.

**Figure 9 biomolecules-10-01624-f009:**
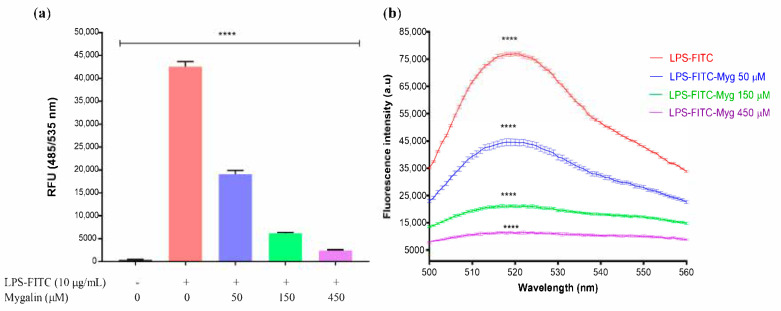
Mygalin binds to LPS. To demonstrate the binding of mygalin to LPS, different concentrations of mygalin were incubated with 10 µg/mL LPS labeled with FITC in saline solution. Mygalin was incubated at 37 °C for 45 min, and then the fluorescence levels were monitored using excitation and a fixed emission (500–560 nm). (**a**) Level of interaction between mygalin and LPS-FITC. (**b**) LPS-FITC fluorescence intensity reduction curve due to mygalin binding measured between 500 and 560 nm. The bars represent the mean ± SEM of four independent experiments. (**** *p* < 0.0001).

**Figure 10 biomolecules-10-01624-f010:**
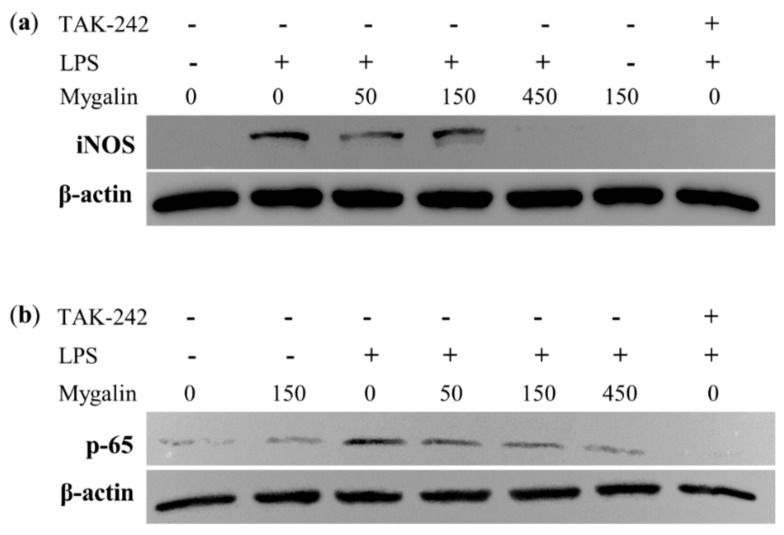
Effects of mygalin on LPS-induced. (**a**) inducible nitric oxide synthase (iNOS) and (**b**) NF-κB p65 protein expression by SDS-PAGE. Western blotting was performed using specific antibodies against iNOS and NF-κB p65 (Cell Signaling). TAK-242 was used as a TLR4 signal inhibitor.

**Figure 11 biomolecules-10-01624-f011:**
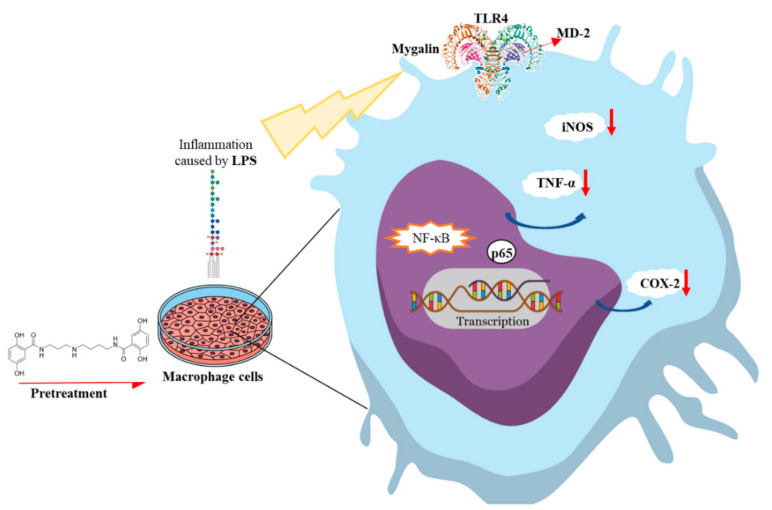
Model of the mechanisms of action of mygalin to inhibit LPS via TLR4. Mygalin binds to the MD-2 protein and blocks the inflammatory effects of LPS.

**Table 1 biomolecules-10-01624-t001:** Primer sequences used in the RT-PCR assay.

Gene	Forward Primer (5′–3′)	Reverse Primer (5′–3′)	Annealing T (°)
iNOS	GCTCCCTATCTTGAAGCCCC	CCCAAACACCAAGCTCATGC	58°C
COX-2	CTTCGGGAGCACAACAGAGT	AATGTTGAAGGTGTCGGGCA	58°C
TNF-α	GTAGCCCACGTCGTAGCAAA	CTGGAAGACTCCTCCCAGGTA	56°C
IL-6	ACGATGATGCACTTGCAGAAAA	GGTCTTGGTCCTTAGCCACTC	56°C
β-actin	TATGCTCTCCCTCACGCCAT	AGGTCTTTACGGATGTCAACG	58°C

**Table 2 biomolecules-10-01624-t002:** Toll-like receptors and other immune factors binding to the ligand mygalin.

Immtorlig_db Compound SMILES	Zinc Database ID	Factors	Tanimoto Similarity Score
CS(=O)(=O)NCCCNC(=O)c1cc2c([nH]c1=O)CCCC2=O	32355850	IL4	0.161
CCNC(=O)NCC(=O)N1CCc2ccccc21	33127803	TLR2/TLR6	0.153
CS(=O)(=O)NCCCNC(=O)c1cc2ccccc2oc1=O	44706237	IL4	0.143
CCNC(=O)CNC(=O)NC(C)(C)C(=O)O	22170390	IL17	0.143
CCS(=O)(=O)NCCCNC(=O)c1cc2ccccc2oc1=O	51461652	IL4	0.141
CCS(=O)(=O)NCCCNC(=O)c1nc(Cl)ccc1Cl	51448237	IL4	0.138
O=C(O)CCCNC(=O)N1CCC[C@H](O)C1	42471780	TLR4/MD-2	0.136
Cc1csc(NC(=O)NCCC(=O)O)n1	40558977	IL17	0.135
CCCNC(=O)NCc1ccc2ccccc2c1	49301164	TLR2/TLR6	0.133
O=C1N=c2ccc(C(=O)NCCC3=CCCCC3)cc2=NC1=O	38776626	IL4	0.133

**Table 3 biomolecules-10-01624-t003:** Molecular properties of the ligands and volume of the binding pocket of TLR4/MD-2.

Drugs	Total Energy (AU)	Solvation Energy (kJ/mol)	Dipole (Debyes)	Polar Surface Area (Å^2^)	Volume (Å^3^)	Area (Å^2^)	TLR4/MD-2
							Binding Pocket Volume (Å^3^)
Mygalin	−1431.66	−94.27	3.41	128.27	417.14	457.35	C = 1187
Curcumin	−1263.16	−49.19	4.21	75.75	373.53	300.12	D = 803
TAK-242	−1895.3	−26.87	4.27	59.93	321.17	344.7	

**Table 4 biomolecules-10-01624-t004:** Interaction energies of the ligands and residues of the binding pocket of the TLR4/MD-2 complexes.

Binding Pocket MD-2-C	Total Free Energy (kJ/mol) TLR4-B/MD-2-C	MD-2-C Residues	Total Residues
**Mygalin**	-236.07	W23, I32, I46, S47, S48, I52, I80, L87, R90, Y131, C133, F151, I153	13
Curcumin	-134.77	I32, I52, L54, I80, R90, I124, F126, Y131, C133, I153	10
TAK-242	-86.99	L54, L78, I80, V82, L87, R90, I124, L125, F126, Y131	10
**Binding Pocket MD-2-D**	**Total Free Energy (kJ/mol)** **TLR4-A/MD-2-D**	**MD-2-D Residues**	**Total Residues**
Mygalin	-259.74	W23, C25, I32, I46, S47, S48, I52, L54, I80, L87, R90, F126, Y131, C133, F151	15
Curcumin	-177.35	I32, I52, L54, I80, L87, R90, I124, F126, Y131, C133, I153	11
TAK-242	-62.94	I80, V82, L87, R90, I124, F126, Y131	7

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
