# Peer review of "Acylpolyamine Mygalin as a TLR4 Antagonist Based on Molecular Docking and In Vitro Analyses"

_biomolecules, 2020, doi:10.3390/biom10121624_

Round 1

Reviewer 1 Report

It is an interesting study and should be helpful. The authors investigated the effects of mygalin on TLR4 in LPS-stimulated murine macrophages by bioinformatics and the in vitro production of immune mediators. These results indicate that mygalin would serve as promising anti-inflammatory potential for future drug development.   

Questions:  

In this study, the author provide only one cell line. It would be better if the author supply another cell line to support these results.

More bioactivity experiments should be performed, such as western blots related to immune mediator should be provided.

Author Response

Comments and Suggestions for Authors

It is an interesting study and should be helpful. The authors investigated the effects of mygalin on TLR4 in LPS-stimulated murine macrophages by bioinformatics and the in vitro production of immune mediators. These results indicate that mygalin would serve as promising anti-inflammatory potential for future drug development.  

Questions: 

1- In this study, the author provide only one cell line. It would be better if the author supply another cell line to support these results.

Reply: Thank you for your analysis and suggestions.

Replay : In this study we work with three cell types, two cell line: J774A.1, RAW 264.7 and macrophages derived from bone marrow (BMDM), which are the primary cells that originate all macrophage populations present in the various organs. Please see items 2.7 (line 152), 2.8 (line 159), 2.9 (line 168), 2.10 (line 174) and 2.11 (line 192) in material and methods.

Also, the methodology was improved, which can be seen in the items highlighted in blue (items 3.4, 3.5, tab.4, Fig.2) referring to line 293 to 323. As well as the discussion line 474-481 and 506-511 and the conclusion on line 597-598.

2- More bioactivity experiments should be performed, such as western blots related to immune mediator should be provided.

Replay: Western blots are not the usual methodology for detecting cytokines. The methods of Elisa, which determines the presence of the active protein and RT-PCR that evaluates the expression of genes, are the methods of choice and were used in our assays. However, for the analysis of enzymes and molecular signals, Western blots were performed as suggested (see MM item 2.15, line 221).

Reviewer 2 Report

In the present article Espinoza-Culupú et al. present mygalin as an agent able to interfere with the inflammation response due to the binding to TLR4. The binding to TLR4 has been studied using docking and several experimental assays demonstrated the inhibition of the inflammation.

The article is not clear in its presentation, the rationale is not well explained and should be better clarified.

My main concern is about the demonstrated binding of mygaling to LPS. This interaction, that has been demonstrated experimentally, could hamper the LPS binding to TLR4/MD-2 and justify the effects observed in 3.7, 3.8 and 3.9, limiting the meaning of the docking studies. To clarify this point some experiment focusing on the direct binding of mygalin to TLR4. excluding the role of LPS is necessary.

Other issues:

  1. Authors define the similarity searching a structure-based virtual screening (paragraph 2.2), but this definition is not correct as a structure-based method, in computer aided drug design, is a method that uses the structure of the target. A similarity searching id a ligand-based method. In this respect, why the Tanimoto thresholds used are different among the diverse databases? Which is the aim of this screening? Unfortunately, I had no access to Supplementary Table 1 that are mentioned in 3.1.
  2. the paragraph 2.6 should be moved to the results section, as it does not present methods but discusses obtained results.
  3. In Table 2 the Tanimoto similarity between mygalin and the other compounds binding immune targets is very low. Is there a reference value for ligands binding suggested by the program or some other reference that can help understanding the meaning of this value?
  4. Paragraph 3.4. Is there some difference between the two dimers?
  5. row 292. the interaction is mediated by one sulfonamide oxygen or by the carbonyl oxygen? A sulfamoyl group does not have carbonyls.
  6. The three ligands have been docked in both the dimeric and tetrameric forms or TLR4 with MD-2. I wonder if the binding to the tetrameric form has a functional meaning.
  7. In the paragraph 3.6 is affirmed “After the in silico analysis of mygalin binding to LPS”. I am quite confused, as the authors presented the binding to TLR4 via MD-2. Could the authors better explain?
  8. The discussion should be revised

Minor issues are related to several misprints:

  • rows 36-38. the sentence between parenthesis should be removed;
  • row 41. “recognize” should be changed in ”they recognize”;
  • row 85, a SMILES string is written without spaces;
  • row 105. “independent docking calculations” remove;
  • row 267. eliminate “great”.
  • row 440. “showed proved”, correct.

Check for other possible errors

Author Response

Comments and Suggestions for Authors

In the present article Espinoza-Culupú et al. present mygalin as an agent able to interfere with the inflammation response due to the binding to TLR4. The binding to TLR4 has been studied using docking and several experimental assays demonstrated the inhibition of the inflammation.

The article is not clear in its presentation, the rationale is not well explained and should be better clarified.

My main concern is about the demonstrated binding of mygaling to LPS. This interaction, that has been demonstrated experimentally, could hamper the LPS binding to TLR4/MD-2 and justify the effects observed in 3.7, 3.8 and 3.9, limiting the meaning of the docking studies. To clarify this point some experiment focusing on the direct binding of mygalin to TLR4. excluding the role of LPS is necessary.

Other issues:

  1. Authors define the similarity searching a structure-based virtual screening (paragraph 2.2), but this definition is not correct as a structure-based method, in computer aided drug design, is a method that uses the structure of the target. A similarity searching id a ligand-based method. In this respect, why the Tanimoto thresholds used are different among the diverse databases? Which is the aim of this screening? Unfortunately, I had no access to Supplementary Table 1 that are mentioned in 3.1.

Reply: Thanks for the suggestion. I apologize for not having accessed the supplementary table in its first revision. I must have made a mistake adding it to the platform. A table and a supplementary figure have been attached for clarification.

The paragraph 2.2 was modified to “Similarity-based virtual screening (VS) of mygalin”

Tanimoto thresholds are different, because each database has a number “N” of molecules registered and differs in quantity, in some cases they are not the same drugs, because of that it is always evaluated in more than one database. For example, ImmtorLig_DB only has 5000 molecules with which the comparison is made.

Supplementary material we have placed at the end of article and also attached

  1. The paragraph 2.6 should be moved to the results section, as it does not present methods but discusses obtained results.

Reply: Thanks for the suggestion.

The paragraph 2.6 was modified to avoid analyzing or describing results and the methods are described with precision.

  1. In Table 2 the Tanimoto similarity between mygalin and the other compounds binding immune targets is very low. Is there a reference value for ligands binding suggested by the program or some other reference that can help understanding the meaning of this value?

Reply: Thanks for your observation.

In the ImmtorLig_DB Article (Figure 8, https://www.nature.com/articles/s41598-018-36179-5) describes the heat map of the ligands (5000) with the experimentally tested molecules available in PubChem where these 5000 ligands had a Tanimoto coefficient of 0.3 according to the analysis, for our mygalin molecule we obtained values between 0.131 and 0.159 half the value obtained by the authors, which would indicate that our molecule could be a possible binder of these receptors despite its low similarity with other Drugs, as mentioned in the ImmtorLig_DB Article in the conclusions.

  1. Paragraph 3.4. Is there some difference between the two dimers?

Reply: Thanks for your observation.

Yes, the volume of the binding pocket (1,187 and 803 A3, table 2). The RMSD of 0.225 A indicates good structural agreement in both complexes, but the superposition shows conformational differences in the side chains of some residues (W23, S47 and I153) of the binding pocket. These differences were reflected in differences in free energy and number of residues bound. However, the overall activity between the three ligands was preserved.

  1. row 292. the interaction is mediated by one sulfonamide oxygen or by the carbonyl oxygen? A sulfamoyl group does not have carbonyls.

Reply: Thanks for your observation.

The interaction is mediated by one sulfonamide oxygen.

The line 320 was corrected:

“while TAK-242 associates with R90 through its sulfonamide oxygen (Figure 2).”

  1. The three ligands have been docked in both the dimeric and tetrameric forms or TLR4 with MD-2.

Reply: Thanks for your observation.

The three ligands have been docked to TLR4 and its coreceptor MD-2. In the script now, we avoid using the name dimer for this complex (TLR4/MD-2).

I wonder if the binding to the tetrameric form has a functional meaning.

Reply: Thanks for your observation.

The aim of the assessment with the tetrameric form was to determine the capability of the ligand to occupy a place in the TLR4/MD-2 complex and once there to evaluate its capability to form an interface stable enough to produce dimerization.

Indeed, the experiment indicated a poor free energy with the tetramer, which supports our proposal. Mygalin occupies the binding domain but is unable to establish a strong enough interaction that could stabilize the tetramer and cause dimerization.

However, we consider that the experiment can be confusing and contrary to the message we are interested in raising, so we have decided not to include the results of the tetramer.

This modification affected the paragraphs:

2.6, 3.5 (deleted), Table 4, Figure 2, and 4. Discussion,

  1. In the paragraph 3.6 is affirmed “After the in silico analysis of mygalin binding to LPS”. I am quite confused, as the authors presented the binding to TLR4 via MD-2. Could the authors better explain?
  2. The discussion should be revised

Reply: Thanks for your observation.

The discussion was revised, and changes were made to the lines:

474 to 488.

  1. Minor issues are related to several misprints:

rows 36-38. the sentence between parenthesis should be removed;

Reply: Thanks for your observation.

Sentence removed

row 41. “recognize” should be changed in ”they recognize”;

Reply: Thanks for the suggestion.

row 38 Changed

row 85, a SMILES string is written without spaces;

Reply: Thanks for your observation.

 row 97, Spaces removed

row 105. “independent docking calculations” remove;

Reply: Thanks for your observation.

 removed

row 267. eliminate “great”.

Reply: Thanks for your observation.

 removed

row 440. “showed proved”, correct.

Reply: Thanks for your observation.

row 454, Proved removed

Check for other possible erros

row 120 sentence “A/C and B/D, where A and B are monomers of TLR4 and C and D are monomers of MD-2” was removed

Other changes were highlighted in blue

The text, after corrected to meet the reviewers' requests, was reedited by the American Journal Expert according to the attached certificate.

Round 2

Reviewer 2 Report

In the revised version of the article the authors addressed most of the issues I raised.

Unfortunately, they didn't answer the most important concern:

"My main concern is about the demonstrated binding of mygaling to LPS. This interaction, that has been demonstrated experimentally, could hamper the LPS binding to TLR4/MD-2 and justify the effects observed in 3.7, 3.8 and 3.9, limiting the meaning of the docking studies. To clarify this point some experiment focusing on the direct binding of mygalin to TLR4 and excluding the role of LPS is necessary."

This point is crucial, in my opinion; otherwise, the computational studies are not confirmed by experimental data.

Author Response

1- Does the introduction provide sufficient. Background and include all relevant references? Can be improved.

Several reorganizations in the text have been carried out and are marked in red throughout the manuscript.

a- summary: lines 26 to 32

b- Introduction: lines 45 to 59 and 94 to 95

2- Is the research design appropriate? Must be improved

The experimental design was reorganized to explain and answer the reviewer's question. Figure 6 was inserted, and the details are in the material and methods in lines 187 to 190 and in the results lines 395 to 402.

3- Are the results clearly presented    Can be improved

The insertion of the new figure 6, explains the question raised by the reviewer, where we show data of the anti-inflammatory effect of Mygalin on the action of LPS, with the cells being pretreated with Mygalin, the drug was removed by washing the culture with medium and later restimulated with LPS. Lines 396 to 409.

4- Are the conclusions supported by the results?  Must be improved

 The discussion and conclusions were reorganized to explain the issues raised. Explanatory text was inserted in the discussion, lines 580 to 582 and 588 to 600. And the conclusion was extended to adapt the new information, lines 648 to 653.

Question:

“My main concern is about the demonstrated binding of mygaling to LPS. This interaction, that has been demonstrated experimentally, could hamper the LPS binding to TLR4/MD-2 and justify the effects observed in 3.7, 3.8 and 3.9, limiting the meaning of the docking studies. To clarify this point some experiment focusing on the direct binding of mygalin to TLR4 and excluding the role of LPS is necessary”.

To clarify this point, an experimental data that had been previously performed in the laboratory was inserted (Fig. 6). In this protocol, J774 cells were pre-incubated with Mygalin for 1 hour and then removed after washing with culture medium. Then "new medium" containing LPS was added and the cells cultured for 24 hrs. There was a reduction in nitrite production, even when stimulation with LPS was performed in the absence of Mygalin. We demonstrate that the action of this molecule can be via interaction with LPS and MD2, confirming in silica assays.

I would like to clarify that a new correction of English will be carried out by the American Journal Expert after the reviewer's decision to avoid successive corrections.